# *Escherichia coli* HS and Enterotoxigenic *Escherichia coli* Hinder Stress Granule Assembly

**DOI:** 10.3390/microorganisms9010017

**Published:** 2020-12-23

**Authors:** Felipe Velásquez, Josefina Marín-Rojas, Ricardo Soto-Rifo, Alexia Torres, Felipe Del Canto, Fernando Valiente-Echeverría

**Affiliations:** 1Molecular and Cellular Virology Laboratory, Virology Program, Faculty of Medicine, Institute of Biomedical Sciences, Universidad de Chile, 8380453 Santiago, Chile; pipe7180@gmail.com (F.V.); josefina.marin@ug.uchile.cl (J.M.-R.); rsotorifo@uchile.cl (R.S.-R.); 2HIV/AIDS Workgroup, Faculty of Medicine, Universidad de Chile, 8380453 Santiago, Chile; 3Microbiology and Mycology Program, Faculty of Medicine, Institute of Biomedical Sciences, Universidad de Chile, 8380453 Santiago, Chile; ale.ntm6@gmail.com (A.T.); felipedelcanto@uchile.cl (F.D.C.)

**Keywords:** stress granules, *E. coli*, integrated stress response

## Abstract

*Escherichia coli*, one of the most abundant bacterial species in the human gut microbiota, has developed a mutualistic relationship with its host, regulating immunological responses. In contrast, enterotoxigenic *E. coli* (ETEC), one of the main etiologic agents of diarrheal morbidity and mortality in children under the age of five in developing countries, has developed mechanisms to reduce the immune-activator effect to carry out a successful infection. Following infection, the host cell initiates the shutting-off of protein synthesis and stress granule (SG) assembly. This is mostly mediated by the phosphorylation of translation initiator factor 2α (eIF2α). We therefore evaluated the ability of a non-pathogenic *E. coli* strain (*E. coli* HS) and an ETEC strain (ETEC 1766a) to induce stress granule assembly, even in response to exogenous stresses. In this work, we found that infection with *E. coli* HS or ETEC 1766a prevents SG assembly in Caco-2 cells treated with sodium arsenite (Ars) after infection. We also show that this effect occurs through an eIF2α phosphorylation (eIF2α-P)-dependent mechanism. Understanding how bacteria counters host stress responses will lay the groundwork for new therapeutic strategies to bolster host cell immune defenses against these pathogens.

## 1. Introduction

In eukaryotic cells, stress granule (SG) assembly can be induced by several environmental stresses including heat shock, oxidative stress, viral infection and UV irradiation, amongst others [1]. SGs are non-membranous cytoplasmic foci, which are formed by mRNA, 40S ribosomal subunits, translation initiator factors (eIF4E, eIF4G, eIF4A, eIF4B, eIF3, eIF2) and RNA-binding proteins such as TIAR, G3BP1 and HuR [2].

The classical mechanism for SG induction is mediated by phosphorylation of the α subunit of the translation initiator factor 2 (eIF2α) by one of the four stress-responsive kinases: HRI, PERK, GCN2 or PKR [3]. Phosphorylation of eIF2α inhibits protein translation by reducing the exchange of the eIF2-GDP to the eIF2-GTP ternary complex necessary for translation initiation (reviewed in [3]). SGs participate in mRNA stabilization and transport, cell survival and cell death [4] and are associated with the pathogenesis of diseases such as cancer, neurodegenerative diseases, inflammatory disorders and viral infections [5]. Although the relationship between viruses and SGs has been widely studied, not much is known about SGs during bacterial infections.

Previous studies have indicated that *Shigella flexneri* affects the ability of infected cells to form canonical SGs and large SG aggregates [6,7]. On the other hand, it has been shown that shiga-toxin-producing *Escherichia coli* is capable of inducing SGs on Caco-2 cells through the subtilase cytotoxin (SubAB) [8]. Here, we compared the ability to induce SGs of two *E. coli* strains, the commensal *E. coli* HS and the enterotoxigenic *E. coli* (ETEC) 1766a, producing the human heat-stable toxin (ST1b), the heat-labile toxin (LT), the coli surface antigen 23 (CS23) and the TleA serine-protease autotransporter [9,10,11]. *E. coli* HS was chosen because it is the prototypic commensal (non-pathogenic) *E. coli* strain, whereas ETEC 1766a was chosen because it is a non-invasive pathogen with a high cell-adherence capacity at short times [10]. Our results indicate that infection of Caco-2 cells with both *E. coli* strains did not induce SG assembly. Furthermore, *E. coli*-infected Caco-2 cells subjected to treatment with sodium arsenite partially inhibit SG assembly through the impairment of eIF2α phosphorylation.

## 2. Materials and Methods

### 2.1. Cell Line and Bacteria Strains

Caco-2 cells (human Caucasian colon adenocarcinoma, obtained from European Collection of Authenticated Cell Cultures, Salisbury, UK) were grown in Dulbecco’s modified Eagle’s medium (DMEM, Gibco Thermo Fisher Scientific, Waltham, Massachusetts, United States) containing 10% fetal bovine serum (FBS, Sigma-Aldrich by Merck KGaA, Darmstadt, Germany) and 1% penicillin–streptomycin (Invitrogen by Thermo Fisher Scientific, Waltham, MA, USA). Cells were grown at 37 °C in a 5% CO_2_ atmosphere. ETEC 1766a and *Escherichia coli* HS was originally isolated from a healthy donor in the US [9,12] and was obtained from the University of Maryland School of Medicine (Baltimore, MD, USA). ETEC 1766a was isolated from a child with watery diarrhea in Santiago, Chile [9,10,11,13].

### 2.2. Bacterial Infection and Stress Treatment

Bacteria were grown in Luria–Bertani (LB) broth (10 g/L tryptone, 5 g/L yeast extract, 5 g/L NaCl) at 37 °C overnight. Caco-2 cells were infected at a multiplicity of infection (MOI) of 100 bacteria/cell at 37 °C in a 5% CO_2_ atmosphere for 1 h. To induce cellular oxidative stress, cells were treated with 0.5 mM sodium arsenite (arsenite, NaAsO_2_; Sigma-Aldrich by Merck KGaA, Darmstadt, Germany) for 1 h or 30 min.

### 2.3. Immunofluorescence and Imaging Analyses

For immunofluorescence (IF) analyses, cells were washed with PBS and fixed with 4% paraformaldehyde for 20 min. Cells were then washed with PBS, incubated in 0.1 M glycine for 10 min, washed with PBS, incubated in 0.2% Triton X-100 for 5 min and washed in PBS. Next, cells were incubated with primary antibodies: anti-G3BP1 mouse (Santa Cruz Biotechnology, Santa Cruz, CA, USA), anti-TIAR rabbit (kindly provided by Dr. Jorge Vera-Otarola, PUC, Chile) for 1 h, and incubated with secondary antibodies anti-mouse 488 or anti-rabbit 594 (Life Technologies) and for the nuclei, cells were stained with DAPI (Sigma-Aldrich by Merck KGaA, Darmstadt, Germany). Confocal microscopy was performed with a Carl Zeiss LSM 700 microscope, and image acquisition was done with a 40× objective lens. All imaging experiments were performed at least 2 times. Imaging analyses were performed using ImageJ software (NIH).

### 2.4. Western Blotting

Cell extracts were prepared by lysis with mild lysis buffer [NaCl 100 mM, Tris-HCl 10 mM pH 7.5, EDTA 1mM, NP40 0.5%, Protease inhibitor cocktail (Roche, 124 CH-4070 Basel Switzerland) and 30 μg of total protein were subjected to 10% SDS-PAGE and transferred to a nitrocellulose blotting membrane (GE Healthcare Boston, MA, USA). Membranes were incubated with an anti-eIF2α (Cell Signaling 32 Tozer Road Beverly, MA, USA), anti-p-eIF2α ser51 (Abcam Cambridge, CB2 0AX, UK) and anti-Actin (Santa Cruz Biotechnology, Santa Cruz, CA, USA). Upon incubation with the corresponding HRP-conjugated secondary antibody (Santa Cruz Biotechnologies, Santa Cruz, CA, USA), membranes were revealed with a C-Digit digital scanner (Li-Cor Lincoln, NE, USA) using the PIERCE^(TM)^ ECL Western blotting substrate (Thermo, Waltham, MA, USA). Signal intensity was quantified by ImageJ (NIH).

### 2.5. Statistical Analysis

The statistical data analysis and graphics described in the text were done using the GraphPad v8.4.3 program (La Jolla, CA, USA). Differences were tested by analysis of variance (Dunnet’s or Bonferroni’s post hoc tests) where *p* < 0.05 was considered significant.

## 3. Results

### 3.1. Non-Pathogenic E. coli HS Strain and ETEC 1766a Pathogenic Strain Do Not Induce Stress Granule Assembly

Our first approach was to determine the capacity of non-pathogenic and pathogenic strains of *Escherichia coli* to induce stress granule assembly in the Caco-2 cells derived from human colorectal adenocarcinoma frequently used to study *E. coli* [14]. We used the non-pathogenic commensal *E. coli* HS strain and the pathogenic enterotoxigenic *E. coli* (ETEC) 1766a strain. To assay the *E. coli*-mediated stress granule assembly, Caco-2 cells were exposed to 0.5 mM of sodium arsenite (Ars) for 30 min or 1 h (as a control for SG assembly) and compared to 1 h of *E. coli* strain infection. Untreated Caco-2 cells, Ars-treated and infected cells were incubated for the indicated times, then fixed and immunostained for the SG marker proteins G3BP1 and TIAR and were stained with DAPI to visualize the nuclei. As shown in Figure 1, untreated cells do not show G3BP1 or TIAR aggregation into dense cytoplasmic foci, consistent with no SG assembly. Ars-treated cells at different times were positive to aggregation and colocalization of G3BP1 and TIAR in cytoplasmic foci. Nevertheless, neither *E. coli* HS nor ETEC strain showed any signs of SG assembly (Figure 1). The presence of *E. coli* strains were proven using DAPI staining and, as expected, bacteria were only observed in the infected cell cultures (Figure 1, lateral panels). These results suggest that *E. coli* infection does not induce SG assembly in Caco-2 cells.

### 3.2. E. coli HS and ETEC 1766a Strains Prevent Arsenite-Induced Stress Granule Assembly in Caco-2 Cells

Several pathogens, including viruses, have developed strategies to subvert the SG response, owing to the fact that SGs constitute an antiviral host response (reviewed in [3,15]). To test if *E. coli* HS and/or ETEC 1766a strains could block SG assembly, we infected Caco-2 cells with *E. coli* HS or ETEC 1766a prior to and after Ars treatment (Figure 2A, left panel). Our results showed that *E. coli* HS- or ETEC 1766a-infected cells treated with Ars were able to prevent the assembly of arsenite-induced SGs in 66% and 58% of cases, respectively (Figure 2A, middle panel in red). Otherwise, when Caco-2 cells were treated with Ars for 30 min and then exposed to *E. coli* HS or ETEC 1766a strains for 30 min, SG assembly was evident. (Figure 2A, middle panel in blue). Finally, we tested the SG assembly upon simultaneous incubation with Ars and *E. coli* HS or ETEC 1766a. Our results showed a reduced SG assembly, 53% and 80%, respectively (Figure 2A middle panel in purple). Although in both conditions (Inf-Ars and Inf + Ars) there is still a low percentage of SG assembly, we observed fewer number of SGs per cell (Figure 2A right panel in red/purple). These results suggest that the presence of *E. coli* strains was sufficient to attenuate the SG assembly mediated by sodium arsenite as a stress inductor (Figure 2B).

### 3.3. E. coli HS and ETEC 1766a Do Not Induce eIF2α Phosphorylation

The canonical mechanism to induce SG assembly is mediated by phosphorylation of eIF2α (eIF2α-P) (reviewed in [3]). Therefore, we evaluated whether the decrease in SG assembly reflected an impairment in eIF2α phosphorylation. Interestingly, the detection of eIF2α-P by Western blot under our different experimental conditions showed that Caco-2 cells subjected to arsenite treatment either post-infection (Inf-Ars) or simultaneous to the infection (Inf + Ars) had a lower signal of eIF2α-P (Figure 2C, in red and purple), which was concomitant with a decrease in SG assembly (Figure 2B). These results indicate that *E. coli* infection prevents SG assembly in response to oxidative stress.

## 4. Discussion

Although some bacterial pathogens can induce SG assembly during infection, it is not clear in each case whether this response benefits the host or the bacteria [16]. Here, we showed that two different *E. coli* strains led to an impairment of stress granule assembly in the epithelial cell line Caco-2, despite the use of sodium arsenite as a stress inductor. Infection times were chosen based on the available literature on *S. flexneri*-induced SG assembly in Caco-2 cells [6,7]. It is worth noting that longer infection times could also bear significant results, like those reported by He et al. [17]. Regardless, our results consistently show that granule assembly impairment was observable at both 30 and 60 min post-infection even in the presence of an exogenous stressor.

Published research showed that the heat-labile enterotoxin from *E. coli* ETEC causes endoplasmic reticulum stress and can induce the phosphorylation of eIF2α in epithelial cells [18]. Given these results, we were surprised to observe that ETEC-infected cells presented impairment in the assembly of SGs and did not show phosphorylation of eIF2α. In contrast, arsenite-induced eIF2α phosphorylation is lower in the presence of bacterial infection. However, it warrants mention that the previous finding was reported on cells treated with the purified toxin, whereas our results were obtained upon directly infecting the cells with the ETEC strain. This implies that the potential impairment of eIF2α phosphorylation could be carried out through another bacterial factor. Similarly, the *E. coli* HS strain had the same reducing effect over oxidative stress-induced eIF2α phosphorylation and SG assembly.

The integrated stress response, including SG assembly, can be heavily modulated during infection by bacterial and cellular factors. The infection’s outcomes remains unclear, as so many variables are involved [16]. In this work, we report a regulatory effect in the cellular stress response, taking place upstream of eIF2α phosphorylation. Given that sodium arsenite induces eIF2α phosphorylation through the HRI kinase, our results suggest that this route is likely targeted [19]. This can be aligned with a recent study that demonstrates that HRI leads the integrated stress response (ISR) and concomitant SG assembly in response to bacterial peptidoglycan detection during infection by intracellular bacteria [20]. Strikingly, the modulation exerted by pathogenic and non-pathogenic *E. coli* strains appears to be similar. This observation could imply that pathogenic bacteria can exploit strategies that allow for commensal bacteria. This could be in part explained by the high average nucleotide identity (ANI) shared by the genomes of the two strains evaluated in this work (98.77%, *E. coli* HS GenBank assembly accession GCA_000017765.1, ETEC 1766a assembly accession GCA_009868995.1). We propose that a lowered stress induction by the commensal *E. coli* HS would favor colonization without upsetting the host. Simultaneously, the ETEC strain could avoid SG induction as an immune-evasion strategy to establish itself on the epithelium, inflicting damage through its repertoire of virulence factors, which are not produced by commensal strains.

## Figures and Tables

**Figure 1 microorganisms-09-00017-f001:**
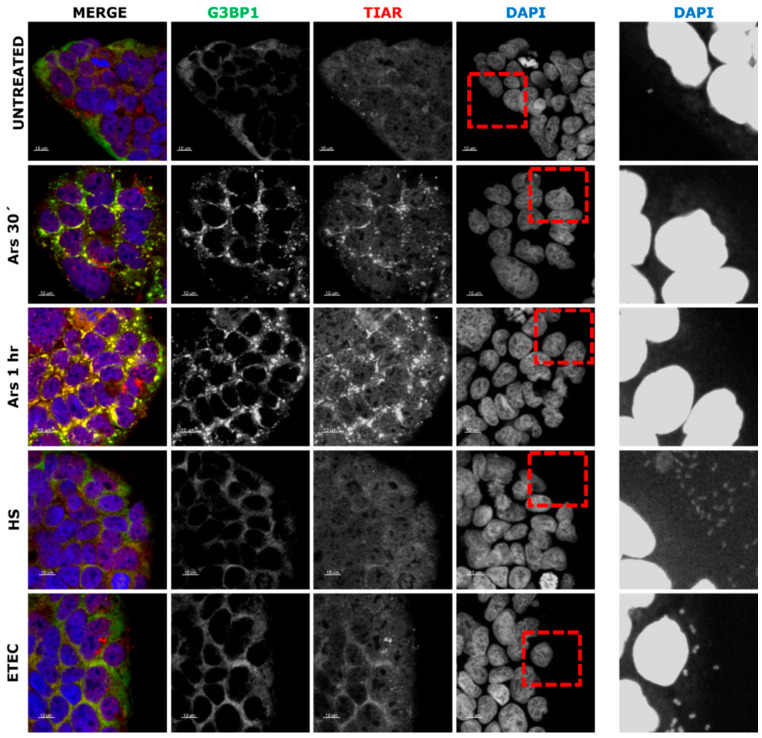
Immunofluorescence (IF) of stress granule marker proteins G3BP1 and TIAR in Caco-2 cells treated with 0.5 mM of sodium arsenite (Ars) at 30 min or 1 h and infected with *E. coli* HS or ETEC 1766a strains. DAPI-stained nuclei and bacteria (right panel). Scale bars are 10 µm.

**Figure 2 microorganisms-09-00017-f002:**
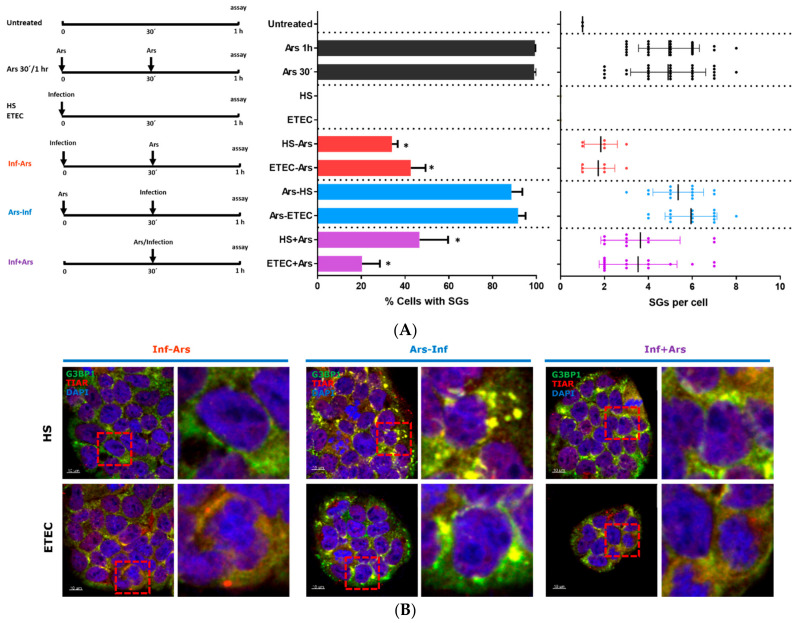
(**A**) Left panel: Schema of the experimental protocol used in this work. The arrows indicate treatment or bacterial infection. Middle panel: Quantification of stress granules (SGs) in Caco-2 cells shown in (B). *p* value is indicated by *(*p* < 0.05 compared with Ars). Right panel: Quantification of SGs per cell from (B). Data are presented as mean ± s.d. from three independent experiments with at least 30 cells analyzed in each condition. (**B**) Immunofluorescence (IF) of stress granule marker proteins G3BP1 and TIAR and nucleus marker DAPI in Caco-2 cells under different experimental conditions. Left panel: Caco-2 cells infected with *E. coli* strains followed by treatment with Ars (Inf-Ars). Middle panel: Cells treated with Ars followed by infection with *E. coli* strains (Ars-Inf). Right panel: Simultaneous *E. coli* infection and Ars treatment (Inf + Ars). Scale bars are 10 µm (**C**) Caco-2 cell lysates were analyzed for eIF2α-p (Ser51), eIF2α and ACTIN, for untreated or infected cells from (B). Densitometry analysis from three independent experiments and representative Western blot are shown. Values were normalized against untreated cells. *p*-value is indicated by * (*p* < 0.05 compared with Ars).

## Data Availability

Data sharing not applicable: No new data were created or analyzed in this study. Data sharing is not applicable to this article.

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
