# Peer review of "Escherichia coli HS and Enterotoxigenic Escherichia coli Hinder Stress Granule Assembly"

_microorganisms, 2020, doi:10.3390/microorganisms9010017_

Round 1
Reviewer 1 Report
Escherichia coli, one of the most abundant bacterial species in the human gut microbiota, have developed a mutualistic relationship with their host, regulating immunological responses. In contrast, enterotoxigenic E. coli (ETEC), one of the main etiologic agents of diarrhea morbidity and mortality in children under 5 in developing countries, has developed mechanisms to reduce its immune-activator effect to carry out a successful infection. Following infection, the host cell initiates the shutting-off of protein synthesis and stress granule (SG) assembly. This is mostly mediated by phosphorylation of translation initiator factor 2α (eIF2α). We therefore have evaluated the ability of a non-pathogenic E. coli strain (E. coli HS) and an ETEC strain (ETEC 1766a) to induce SG assembly in response to exogenous stresses. We found that infection of Caco-2 cells with both E. coli strains did not induce SG assembly on their own, but did partially inhibit this assembly in cells pretreated with sodium arsenite through impairment of eIF2α phosphorylation. Understanding how bacteria counter host stress responses promises to lay the groundwork for new therapeutic strategies to bolster host cell immune defenses against these pathogens. (Modified after the Abstract)
Formation of stress granules (SG) by eukaryotic cells is an evolutionarily conserved response to environmental stresses including those posed by pathogens. As the authors note, the relationship between SG and viruses has been extensively studied but studies of SG and other pathogens such as bacteria are few. The current study is thus an important addition to the literature. In contrast to results obtained by others with Shigella flexneri, the authors found that while non-pathogenic and pathogenic E. coli did not directly induce SG in infected Caco cells in culture they did prevent SG assembly in cells undergoing oxidative stress in response to arsenite and involved an eIF2α-dependent mechanism. The implications of these findings are well discussed by the authors. Unfortunately, these strengths are undercut by several easily remediable editorial problems.
- Abstract (lines 20-23) and Introduction (lines 46-48). These ideas much better expressed in the last lines of the Introduction than in the Abstract. The Abstract would therefore be easier to understand by a casual reader if it were revised accordingly.
- Introduction. There are so few articles dealing with SG in bacterial infections that the authors might consider adding a citation to Vonaesch et al.. J Vis Exp. (125):55536, 2017.
- Materials and Methods. The sources of the E. coli strains and the sodium arsenite should be indicated.
- line 145. Revising this sentence to indicate “that E. coli infection prevents” SG assembly or formation “in response to oxidative stress” would make its meaning easier for the reader to understand.
- Other minor changes. Line 14, Escherichia should be abbreviated since it was written in full in the previous sentence. Line 14, "diarrhea” should be changed to “diarrheal.” Line 15, “years” is not necessary and should be omitted. Line 84, “does” should be changed to “do”.
Author Response
Reply:
We want to thank the reviewer for the careful and thorough reading of this manuscript and the thoughtful comments and constructive suggestions, which help improve our manuscript's quality.
Comments:
- Abstract (lines 20-23) and Introduction (lines 46-48). These ideas much better expressed in the last lines of the Introduction than in the Abstract. The Abstract would therefore be easier to understand by a casual reader if it were revised accordingly.
Response: As suggested by the reviewer, we have rewritten lines 20-23 from the abstract and 46-48 from the introduction to improve the paragraph.
- There are so few articles dealing with SG in bacterial infections that the authors might consider adding a citation to Vonaesch et al.. J Vis Exp. (125):55536, 2017.
Response: We have added the reference
- Materials and Methods. The sources of the colistrains and the sodium arsenite should be indicated.
Response: We have incorporated the information requested.
- line 145. Revising this sentence to indicate “that coliinfection prevents” SG assembly or formation “in response to oxidative stress” would make its meaning easier for the reader to understand.
Response: We have rewritten it to make it more consistent.
- Other minor changes. Line 14, Escherichiashould be abbreviated since it was written in full in the previous sentence. Line 14, "diarrhea” should be changed to “diarrheal.” Line 15, “years” is not necessary and should be omitted. Line 84, “does” should be changed to “do”.
Response: Thanks for your detailed review. The changes have been made.
Reviewer 2 Report
Introduction section is rather short and in the end of this part there are two sentences with information about the result. It should be moved to the results section. Instead, Authors may give information why they choose E. coli HS and enterotoxigenic E. coli.
Additionally, the legend od Fig. 2 is not very friendly for reading. Maybe the Authors could improve it somehow.
In Discussion please give the names of authors of references no. 14 before the brackets.
I would also recommend giving more precise information about Authors Contribution.
Author Response
Response to Reviewer #2:
We want to thank the reviewer for the careful and thorough reading of this manuscript and the thoughtful comments and constructive suggestions, which help improve our manuscript's quality.
Comments and Suggestions for Authors
Introduction section is rather short and in the end of this part there are two sentences with information about the result. It should be moved to the results section. Instead, Authors may give information why they choose E. coli HS and enterotoxigenic E. coli.
Response: As suggested by the reviewer, we have rewritten the introduction to improve the paragraph and we have incorporated the information requested.
Additionally, the legend od Fig. 2 is not very friendly for reading. Maybe the Authors could improve it somehow.
Response: We have rewritten it to make it more friendly to the public.
In Discussion please give the names of authors of references no. 14 before the brackets.
Response: Thank you for pointing this out. We have corrected it properly.
I would also recommend giving more precise information about Authors Contribution.
Response: We have incorporated your suggestion.